# Chondroprotective Effects of a Histone Deacetylase Inhibitor, Panobinostat, on Pain Behavior and Cartilage Degradation in Anterior Cruciate Ligament Transection-Induced Experimental Osteoarthritic Rats

**DOI:** 10.3390/ijms22147290

**Published:** 2021-07-07

**Authors:** Zhi-Hong Wen, Jhy-Shrian Huang, Yen-You Lin, Zhi-Kang Yao, Yu-Cheng Lai, Wu-Fu Chen, Hsin-Tzu Liu, Sung-Chun Lin, Yu-Chi Tsai, Tsung-Chang Tsai, Yen-Hsuan Jean

**Affiliations:** 1Department of Marine Biotechnology and Resources, National Sun Yat-sen University, Kaohsiung 80424, Taiwan; wzh@mail.nsysu.edu.tw (Z.-H.W.); akang329@vghks.gov.tw (Z.-K.Y.); D52449@auh.org.tw (Y.-C.L.); ma4949@adm.cgmh.org.tw (W.-F.C.); 2Section of Orthopedics, Department of Surgery, Antai Medical Care Corporation Anti Tian-Sheng Memorial Hospital, PingTong 92842, Taiwan; jshuang9208@gmail.com; 3Department of Sports Medicine, China Medical University, No. 91 Hsueh-Shih Road, Taichung 40402, Taiwan; chas6119@gmail.com; 4Department of Orthopedics, Kaohsiung Veterans General Hospital, Kaohsiung 81341, Taiwan; 5Department of Orthopedics, Asia University Hospital, Taichung 41354, Taiwan; 6Department of Neurosurgery, College of Medicine, Kaohsiung Chang Gung Memorial Hospital and Chang Gung University, Kaohsiung 83301, Taiwan; 7Department of Medical Research, Hualien Tzu Chi Hospital, Buddhist Tzu Chi Medical Foundation, Hualien 97002, Taiwan; hsintzu_liu@tzuchi.com.tw; 8Department of Orthopedic Surgery, Pingtung Christian Hospital, No. 60 Dalian Road, Pingtung 90059, Taiwan; 02572@ptch.org.tw; 9National Museum of Marine Biology and Aquarium, Pingtung 94450, Taiwan; yuchi0713@gmail.com; 10Section of Nephrology, Department of Medicine, Antai Medical Care Corporation Anti Tian-Sheng Memorial Hospital, Pingtung 92842, Taiwan; n833_d10@yahoo.com.tw

**Keywords:** histone deacetylases, panobinostat, osteoarthritis, nociception

## Abstract

Osteoarthritis (OA) is the most common articular degenerative disease characterized by chronic pain, joint inflammation, and movement limitations, which are significantly influenced by aberrant epigenetic modifications of numerous OA-susceptible genes. Recent studies revealed that both the abnormal activation and differential expression of histone deacetylases (HDACs) might contribute to OA pathogenesis. In this study, we investigated the chondroprotective effects of a marine-derived HDAC inhibitor, panobinostat, on anterior cruciate ligament transection (ACLT)-induced experimental OA rats. The intra-articular administration of 2 or 10 µg of panobinostat (each group, *n* = 7) per week from the 6th to 17th week attenuates ACLT-induced nociceptive behaviors, including secondary mechanical allodynia and weight-bearing distribution. Histopathological and microcomputed tomography analysis showed that panobinostat significantly prevents cartilage degeneration after ACLT. Moreover, intra-articular panobinostat exerts hypertrophic effects in the chondrocytes of articular cartilage by regulating the protein expressions of HDAC4, HDAC6, HDAC7, runt-domain transcription factor-2, and matrix metalloproteinase-13. The study indicated that HDACs might have different modulations on the chondrocyte phenotype in the early stages of OA development. These results provide new evidence that panobinostat may be a potential therapeutic drug for OA.

## 1. Introduction

Panobinostat (LBH589), a histone deacetylase (HDAC) inhibitor, was approved for patients with multiple myeloma and other hematological malignancies by the United States Food and Drug Administration (US-FDA) in 2015 [1]. LBH589 was developed from a marine natural product, psammaplin A (PSA) [2,3,4], which was first isolated from a marine sponge *Psammaplvsilla* sp. in 1987 [5]. At first, PSA was discovered as an HDAC inhibitor for anticancer properties from p21, a cyclin-dependent kinase 2 promoter assay system [6,7]. Due to the weak physiological stability of PSA, medicinal chemists attempted to improve its chemical properties [8]. Although several interesting derivatives were developed, the pharmacological profile of PSA was not improved successfully [7,8]. Subsequently, another natural compound, trichostatin A (also an HDAC inhibitor), was isolated using the same p21 promoter assay [7,8,9]. LAK974 was synthesized based on the analysis of a two-dimensional pharmacophore model of trichostatin A; however, LAK974 showed significant activity in vitro but poor efficacy in vivo [7]. Structure modification was continued by computational docking studies to obtain LAQ824, which showed increased antiproliferative effects in several cancer cell lines (including A549 (lung), HCT166 (colon), and MDA435 (breast)) and decreased apoptotic death in normal human dermal fibroblasts [7]. As the safety of LAQ824 in preclinical development was unclear and induced a body weight loss problem in vivo [7], LBH589 was finally optimized by further synthetic design and showed a significant tumor regression effect in an HCT116 xenograft model with minimal body weight loss [7,8].

HDACs, a class of epigenetic factors, epigenetically regulate chromatin structure and transcription factor activity by removing acetyl groups from lysine residues on histones [10,11,12]. Recent studies have demonstrated that the abnormal activation and differential expression of HDAC contribute to osteoarthritis (OA) pathogenesis [10,11,12] and suggested that inhibiting or augmenting the activity of specific HDACs plays an important role in controlling OA development and progression. However, recent intensive studies have demonstrated that the chondroprotective effects of HDAC inhibition attenuate matrix metalloproteinases (MMPs) and consequently reduce cartilage degradation [10,11,12]. Thus, HDAC inhibitors might emerge as a promising new class of therapeutic drug for OA in recent years. Several studies indicated that panobinostat has beneficial effects in the downregulation of proinflammatory cytokines [13,14,15]. In addition, panobinostat modulates micro-RNA-146a, a negative regulator of cytokine-induced inflammation response, expression in OA fibroblast-like synoviocytes [14]. The authors proposed that panobinostat might have potential therapeutic effects for OA. The aforementioned results suggest that modulating HDAC activity by panobinostat offers a potential treatment option to prevent the molecular events involved in OA pathogenesis.

OA is a common age-related degenerative joint disease that causes pain and disability in older people. It is characterized not only by cartilage degradation, subchondral bone remodeling, and synovial inflammation, but also by inflammation, infrapatellar fat pad fibrosis, and meniscal damage/degeneration [16,17,18]. Epidemiological studies have estimated the prevalence of OA to be approximately 3.8% in the global population, increasing with age, and that it generally affects women more frequently than men [19,20,21]. Thus, old age is an evident risk factor for OA development. The estimated incidence of OA varies greatly across regions and populations depending on its definition, suggesting the roles of lifestyle and environmental factors in its etiology [18,22]. Moreover, the trend in OA incidence has increased over recent decades and is still increasing in Western and Asian countries [21,23]. Due to the lack of disease-modifying treatments for patients with OA, OA consumes a substantial amount of healthcare resources; this results in immense socioeconomic burden in developed countries [24,25]. Therefore, there is still an urgent clinical need to develop efficient and cost-effective approaches to alleviate OA development and progression.

Articular cartilage consists of a dense extracellular matrix (ECM) containing embedded distributed chondrocytes. Chondrocyte is the only cell type that is resident in articular cartilage and is responsible for maintaining the structural and functional integrity of the cartilage matrix [26,27]. The degradative process of the cartilage matrix is primarily mediated by the excessive production of ECM-degrading enzymes, such as MMPs, to cleave various ECM components [28,29]. Within the MMP family, MMP13 exhibits high activity, abundantly expresses protease in OA chondrocytes, and serves as an essential protease involved in cartilage degradation [28,29]. Runt-domain transcription factor-2 (RUNX2), a critical transcription factor for chondrocyte hypertrophy [30], was shown to enhance MMP13 promotor activity and thereby promote cartilage degradation [31,32]. A study showed that RUNX2 deletion decelerates OA progression in an experimental OA model using medial meniscal destabilization surgery [33]. These findings suggest that targeting RUNX2 and its downstream effector MMP13 could be potential therapeutic strategies for OA treatment.

In the present study, we evaluated the in vivo effects of panobinostat in OA development using a rat experimental model of anterior cruciate ligament (ACL) transection (ACLT)-induced OA. We examined nociceptive behaviors, including secondary mechanical allodynia and weight-bearing distribution, after intra-articularly injecting panobinostat. To evaluate the morphometric change of the joint structure, microcomputed tomography (CT) and histopathological analyses were performed to observe joint destructions. Furthermore, immunohistochemical staining was performed to clarify the role of HDAC4, HDAC6, and HDAC7 as well as the chondrocyte hypertrophic marker RUNX2.

## 2. Results

### 2.1. Panobinostat Attenuates ACLT-Induced OA Progression

To investigate the potential effects of panobinostat on reducing pain in OA progression, we intra-articularly administered two dosages (2 and 10 μg as low and high doses, respectively) of panobinostat into rats after ACLT. Hind paw weight distribution (Figure 1A) and secondary mechanical allodynia (Figure 1B) were measured as pain-related behavior assessments. Compared with the control group, changes in the hind paw weight distribution had increased significantly (64.7 ± 5.7 vs. 2.2 ± 1.1 g; *p* < 0.01; Figure 1A), and the paw withdrawal threshold had decreased significantly (1.2 ± 0.2 vs. 10.0 ± 0.0 g; *p* < 0.01; Figure 1B) at the 24th week after ACLT. These results indicate that weight-bearing asymmetry and mechanical allodynia were produced with ALCT-induced OA progression. In the weight-bearing distribution test, the inhibitory effects of panobinostat were associated with a dose-dependent reduction in the groups of ACLT + panobinostat (2 and 10 μg) treatments. The weight-bearing distribution was significantly lower in the ACLT with 10 μg of panobinostat group than in the ACLT group throughout the period of panobinostat administration at the 6th–17th weeks (65.67 ± 3.89 vs. 59.06 ± 4.41 g; 15.54 ± 3.32 vs. 68.6 ± 4.18 g; 16.9 ± 4.39 vs. 54.4 ± 2.6 g; 5.3 ± 2.18 vs. 56.29 ± 3.54 g; 5.71 ± 2.14 vs. 56.7 ± 2.82 g; 4.91 ± 2.16 vs. 61.06 ± 3.08 g, respectively; *p* < 0.05; Figure 1A). The same trend was observed in the comparison between the ACLT with 2 μg panobinostat and ACLT groups at the 6th–17th weeks (56.86 ± 4.91 vs. 59.06 ± 4.41 g; 32.71 ± 3.54 vs. 68.6 ± 4.18 g; 20.07 ± 2.98 vs. 54.4 ± 2.6 g; 23.93 ± 3.7 vs. 56.29 ± 3.54 g; 25.28 ± 4.38 vs. 56.7 ± 2.82 g; 22.45 ± 2.52 vs. 61.06 ± 3.08 g, respectively; *p* < 0.05; Figure 1A). Moreover, we observed that the inhibitory effects persisted after stopping the administration of panobinostat up to the 24th week after ACLT (45.48 ± 85 vs. 56.29 ± 3.54 g; 45.48 ± 2.86 vs. 56.29 ± 3.54 g; *p* < 0.05; Figure 1A). During the period of panobinostat administration at the 6th–17th weeks after ACLT, the paw withdrawal threshold was significantly increased in a dose-dependent manner compared with the ACLT group (Figure 1B). As shown in Figure 1C, the ACLT group experienced a significantly increased swelling of the hind limb knee joint before panobinostat treatment. The width of the hind limb knee joint significantly decreased in the ACLT with 10 μg of panobinostat group than in the ACLT group throughout the period of panobinostat administration at the 6th–17th weeks (0.79 ± 0.1 vs. 0.73 ± 0.06 mm; 0.36 ± 0.05 vs. 0.82 ± 0.05 mm; 0.25 ± 0.04 vs. 1.02 ± 0.05 mm; 0.25 ± 0.04 vs. 1.01 ± 0.03 mm; 0.37 ± 0.05 vs. 0.99 ± 0.05 mm; 0.41 ± 0.06 vs. 1.02 ± 0.04 mm, respectively; *p* < 0.05; Figure 1C). The same trend was noted in the comparison between the ACLT with 2 μg of panobinostat and ACLT groups at the 6th–17th weeks (0.79 ± 0.06 vs. 0.73 ± 0.06 mm; 0.34 ± 0.04 vs. 0.82 ± 0.05 mm; 0.35 ± 0.07 vs. 1.02 ± 0.05 mm; 0.34 ± 0.16 vs. 1.01 ± 0.03 mm; 0.4 ± 0.03 vs. 0.99 ± 0.05 mm; 0.51 ± 0.08 vs. 1.02 ± 0.04 mm; *p* < 0.05; Figure 1C).

No significant differences were observed in body weight between the experimental groups even during panobinostat administration (Figure 1D). Taken together, these results suggest that panobinostat treatment effectively alleviates pain and reduces inflammation in ACLT-induced OA rats.

### 2.2. Micro-CT Analysis of the Effect of Panobinostat on ACLT-Induced OA Rat

The three-dimensional images by micro-CT were reconstructed to examine the effects of panobinostat on the structural changes of the knee after ACLT surgery. Representative three-dimensional micro-CT images of the subchondral bones and their microarchitectural-related parameters are presented in Figure 2. The knee joints of the ACLT group showed a rougher and more irregular surface than those of the control rats (Figure 2A). Meanwhile, panobinostat treatment prominently reduced ACLT-induced osteophyte formation (Figure 2A). In addition, there was no visible bone erosion on the subchondral plate in the sagittal views of the control group. Compared with the control group, the bone surface was significantly higher in the ACLT group (*p* < 0.05; Figure 2B). By contrast, Tb.N (Figure 2C) and bone mineral density (BMD; Figure 2D) were significantly lower in the ACLT group (*p* < 0.05). The ACLT with panobinostat groups had significantly reduced bone surface (*p* < 0.05; Figure 2B) and increased Tb.N (Figure 2C) and BMD (Figure 2D) than the ACLT only group (*p* < 0.05). The changes in histomorphometric parameters suggest that panobinostat treatment predominantly attenuates subchondral remodeling in ACLT-induced OA progression. 

### 2.3. Panobinostat Attenuates Cartilage Degradation in ACLT-Induced OA Model

To evaluate the preventive effects of panobinostat treatment on cartilage degradation, histological analysis using Safranin O/Fast Green staining was performed in the tissue sections of knee joints from the control, ACLT, and ACLT + panobinostat (2 or 10 μg) groups. The image of the control group showed a regular morphological structure in articular cartilage. Compared with the control group, cartilage superficial destruction and cartilage erosion were observed in the ACLT group. Compared with the ACLT group, intra-articular injection with panobinostat (2 or 10 μg) from the 6th to 17th week reduced cartilage and bone erosion (Figure 3A). Cartilage histopathology was further assessed using the Osteoarthritis Research Society International (OARSI) histological scoring system by Safranin O/Fast Green staining. The OARSI score was significantly higher in the ACLT group (12.8 ± 1.6) than in the control group (0.2 ± 0.2; *p* < 0.05; Figure 3B). In contrast with the ACLT group, panobinostat administration (2 or 10 μg) led to significantly lower OARSI scores (5.7 ± 0.8 vs. 12.8 ± 1.6; 4.8 ± 0.7 vs. 12.8 ± 1.6, respectively; *p* < 0.05; Figure 3B). There was no significant difference in the OARSI scores among the panobinostat groups. These results suggest that panobinostat administration attenuates cartilage degradation in ACLT-induced OA rats.

### 2.4. Panobinostat Affects HDAC4, HDAC6, and HDAC7 Expressions in an ACLT-Induced OA Model

The immunohistochemical staining of HDAC4, HDAC6, and HDAC7 revealed that panobinostat ameliorates ACLT-induced cartilage damage progression. Figure 4A illustrates the distribution of HDAC4-, HDAC6-, and HDAC7-positive cells in the cartilage tissues of knee joints from the control, ACLT, and ACLT + panobinostat (2 or 10 μg) groups. The quantitative analysis of immunohistochemical staining showed that HDAC4-positive cells were significantly downregulated in the ACLT group (16.2 ± 1.2) than in the control group (33.6 ± 2.5; *p* < 0.05; Figure 4B). The ACLT + panobinostat (2 or 10 μg) groups reversed the ACLT-induced downregulation of HDAC4-positive cells in a dose-dependent manner (24.0 ± 2.7 or 36.2 ± 3.3; Figure 4B). Compared with the control group, the numbers of HDAC6- and HDAC-7-positive cells were significantly upregulated in the ACLT group (29.4 ± 2.0 vs. 13.3 ± 1.7; 33.2 ± 2.3 vs. 6.5 ± 0.9, respectively; *p* < 0.05; Figure 4C,D). The ACLT + panobinostat groups showed an attenuated ACLT-induced upregulation of HDAC6- (9.6 ± 1.1 vs. 29.4 ± 2.0; 9.5 ± 0.7 vs. 29.4 ± 2.0, respectively; *p* < 0.05; Figure 4C) and HDAC7-positive (11.9 ± 1.5 vs. 33.2 ± 2.3; 8.29 ± 1.6 vs. 33.2 ± 2.3, respectively; *p* < 0.05; Figure 4D) cell numbers in chondrocytes. These results suggest that panobinostat administration significantly upregulates HDAC4 and downregulates HDAC6 and HDAC7 in cartilage tissues after ACLT.

### 2.5. Panobinostat Inhibits Articular Cartilage Hypertrophy

Chondrocyte hypertrophy, regulated by RUNX2 and its downstream effector MMP13, is a critical biological event that involves cartilage degeneration during OA pathogenesis [30]. The immunohistochemical staining of RUNX2 and MMP13 protein expressions were used to examine the effects of panobinostat on chondrocyte hypertrophy in ACLT-induced OA. The distribution of RUNX2- and MMP13-positive cells in the cartilage tissues of knee joints from the control, ACLT, and ACLT + panobinostat (2 or 10 μg) groups is presented in Figure 5A. Compared with the control group, the quantitative analysis of immunohistochemical staining showed that RUNX2- and MMP13-positive cells were significantly higher in number after ACLT surgery (52.1 ± 3.6 vs. 12.2 ± 1.3; 78.2 ± 3.6 vs. 4.7 ± 1.4, respectively; *p* < 0.05; Figure 5B,C). However, both low- and high-dose panobinostat treatments markedly decreased the number of RUNX2- (35.7 ± 5.4 vs. 52.1 ± 3.6; 30.7 ± 3.6 vs. 52.1 ± 3.6, respectively; *p* < 0.05; Figure 5B) and MMP13-positive cells compared with the ACLT group (58.4 ± 1.9 vs. 78.2 ± 3.6; 46.9 ± 3.3 vs. 78.2 ± 3.6, respectively; *p* < 0.05; Figure 5C). There was no significant difference between the low- and high-dose panobinostat treatment groups (Figure 5B,C). These results suggest that panobinostat administration significantly downregulates RUNX2 and MMP13 in cartilage tissues after ACLT.

## 3. Discussion

In the present study, we determined the therapeutic effect of a marine-derived HDAC inhibitor, panobinostat, by its intra-articular injection in the ACLT-induced OA rat model. To the best of our knowledge, this is the first study to investigate and show the protective effects of panobinostat in an experimental OA animal model. Our results clearly showed that intra-articular panobinostat could attenuate OA development at an early stage and effectively ameliorate nociceptive behaviors, including secondary mechanical allodynia and weight-bearing distribution. Moreover, panobinostat exerts the protective effects of articular cartilage degradation and subchondral bone changes after ACLT by micro-CT analysis and OARSI cartilage histopathology assessment. We also described a dual chondroprotective effect of panobinostat on articular cartilage degradation and chondrocyte hypertrophy via class II HDAC family protein modulations, including HDAC4, HDAC6, and HDAC7 expressions. The intra-articular HDAC inhibitor, panobinostat, also attenuated RUNX2 and its downstream effector MMP13 in the chondrocytes of cartilage.

Persistent and chronic joint pain is the predominant clinical feature of OA. It is known that chondrocytes respond to the accumulation of injurious biochemicals and biomechanical insults acquired from a hypertrophy-like phenotype, which plays a vital role in the onset and development of OA [27,34]. The murine models of OA have been used to study joint pain that recapitulates disease manifestations similar to human OA [35]. In the present study, mechanical allodynia and hind paw weight distribution were measured to assess the response of pain in the ACLT-induced OA model. The intra-articular administration of panobinostat attenuated mechanical allodynia threshold and improved weight-bearing distribution from the 6th to 17th weeks after ACLT, suggesting that HDACs participate in OA-induced ongoing nociception. Furthermore, inflammatory changes concurrent with the ACLT-induced model included joint effusion and synovial membrane hyperplasia, which represent the important mechanisms of joint pain in patients with OA [36]. In such patients, changes in knee joint width could be measured to determine the extent of tissue swelling as an index of inflammation [37]. Our results showed that the ACLT + panobinostat groups had significantly reduced knee joint width compared with the ACLT only group, suggesting that the intra-articular injection of panobinostat decreases inflammation in the ACLT knee. HDAC inhibitors are most commonly used as anticancer drugs. Several studies have also indicated the therapeutic potential of HDAC inhibitors as anti-inflammatory and immunosuppressive agents [38,39,40]. Moreover, they are effective analgesics [41]. OA is a type of degenerative joint disease that accompanies cartilage degeneration and continuous nociception. Pathological inflammation plays a vital role in the development and progression of cartilage degeneration and nociceptive sensitization in OA [42]. Moreover, RUNX2 and MMP13 are involved in inflammation and cartilage destruction in OA [43,44,45]. Both cartilage degradation and nociceptive sensitization can be relieved by MMP13 inhibition [46,47,48]. MMP13 is specifically expressed in the cartilage of patients with OA but not in normal adult cartilage [49,50]. In the present study, panobinostat significantly inhibited ACLT-induced RUNX2 and MMP13 protein expression (Figure 5). Therefore, we consider panobinostat to be of therapeutic value for OA via its anti-inflammatory properties by MMP13 inhibition.

Once OA is initiated, abnormal hypertrophic chondrocytes in articular cartilage produce more catabolic factors involved in cartilage degradation that ultimately result in ECM degradation and, consequently, in progressive joint degeneration [26,34]. Cumulative studies support the concept that the progressive loss of articular cartilage and acceleration of subchondral bone turnover lead to microarchitecture changes in the subchondral trabecular bone of OA joints, which is characterized by increased subchondral plate thickness and osteophyte formation [51,52]. Changes in the subchondral bone microarchitecture related to articular cartilage degeneration have been reported in human OA knee and several experimental OA models [53,54]. Our previous studies clearly indicated that ACLT-induced nociceptive sensitization was highly correlated with higher OARSI histological scores, which reflect pathological changes in rat cartilages [55,56]. The dysregulation between anabolic and catabolic factors is a crucial event in articular cartilage degradation in OA. In the present study, histopathological observations demonstrated that the OARSI score was significantly lower in the ACLT + panobinostat groups than the ACLT only group. This finding supports a protective role of panobinostat in ACLT-induced joint inflammation and cartilage degradation, suggesting that panobinostat alleviates clinical signs and retards OA progression. Furthermore, the present study examined the chondroprotective effect of panobinostat on subchondral bone quality to evaluate the relationship between subchondral bone and cartilage degeneration. Micro-CT, a well-established and validated technique, provides a quantitative and nondestructive three-dimensional imaging modality that has been used to analyze the microarchitecture of subchondral bone [57,58]. In the present study, the micro-CT images displayed bone erosion on the surfaces of the subchondral plate after ACLT treatment, which was alleviated by panobinostat administration. In addition, panobinostat treatment prominently reduced ACLT-induced osteophyte formation. The trabecular bone scanned by micro-CT revealed a significant increase in the bone surface and a significant decrease in BMD and Tb.N in the ACLT group. The intra-articular administration of panobinostat significantly decreased bone surface and significantly increased BMD and Tb.N after ACLT. Further, the micro-CT analysis revealed an aberrant subchondral bone formation in the ACLT-induced OA group, which was abrogated by panobinostat treatment. This finding suggests that panobinostat attenuates OA subchondral bone remodeling with less osteophyte formation. Taken together, we provided a hypothesis that panobinostat indirectly reduces cartilage degradation by protecting subchondral bone from resorption. 

HDAC proteins are grouped into classes I–IV based on DNA sequence similarity and activities [10,11]. Several HDACs are expressed in chondrocytes both in normal cartilage and the diseased cartilage of patients with OA [10,11,59]. It has been suggested that different HDACs may have a different impact on chondrocyte phenotype. HDAC4 expression is significantly negatively correlated with OA severity [60], which is consistent with our observation that HDAC4-positive cells are significantly diminished in the ACLT group compared with the control group. Previous studies have shown that RUNX2 and its downstream effector MMP13 are both upregulated at the early stage of OA [43,44] and simultaneously observed in OA human chondrocytes and OA animal models [44,45]. HDAC4 downregulation in SW1353 chondrocyte-like cells transfected with HDAC4-specific siRNA resulted in increased RUNX2 and MMP13 expressions [61]. By contrast, HDAC4 upregulation in rat chondrocytes transduced with HDAC4 adenoviral vector reduced RUNX2 and MMP13 expression [62]. HDAC4 is a negative regulator of chondrocyte hypertrophy because it suppresses several chondrocyte-hypertrophy-related genes, including RUNX2 and MMP13, by inhibiting their promotor activities [60,63]. Thus, HDAC4 upregulation might exhibit a chondroprotective effect by inhibiting RUNX2 and MMP13 transcription activities. Consistent with these findings, our data showed that the number of HDAC4-positive cells significantly increased in a dose-dependent manner in the ACLT + panobinostat group than in the ACLT only group. Meanwhile, we found that the number of RUNX2- and MMP13-positive cells significantly decreased in the ACLT + panobinostat groups. The therapeutic effect of HDAC4 in vivo has been examined in the ACLT-induced OA rat model by an intra-articular injection of an adenoviral vector containing HDAC4 into the articular cartilage of the knee [62]. In this study, Gu et al. indicated that HDAC4 upregulation effectively attenuated articular cartilage damage by RUNX2 and MMP13 repression, subsequently reducing osteophyte formation and cartilage damage, and increasing articular cartilage anabolism [62]. This finding demonstrated that HDAC4 had a chondroprotective effect, which further supports the reliability of our results that the panobinostat-mediated upregulation of HDAC4 could slow disease progression during the early stages of OA.

In contrast to HDAC4 downregulation, elevated expressions of HDAC6 and HDAC7 have been observed in the cartilage tissues of patients with OA [64,65]. This is in line with our observations that HDAC6- and HDAC7-positive cells are significantly increased in the ACLT group compared to the control group. Elevated HDAC7 expression in human OA may contribute to cartilage degradation by promoting MMP13 expression [65], whereas HDAC7 downregulation by miR-193b-5p inhibits MMP13 expression to reduce cartilage degradation [66]. Li et al. found that ricolinostat (ACY-1215), a selective HDAC6 inhibitor, inhibits MMP13 expression in the articular cartilage and prevents cartilage degradation in OA mice [64]. Consistent with these results, our data showed that the number of HDAC6- and HDAC7-positive cells significantly decreased after panobinostat treatment. This evidence revealed that HDAC6 and HDAC7 inhibition could prevent cartilage degradation, which further supports the reliability of our results that panobinostat mediated HDAC6 and HDAC7 downregulation. Based on the aforementioned results, the multiple functions and targets of class II HDAC might play different roles via different mechanisms at the early stages of OA development. Recently, panobinostat was confirmed to inhibit the enzymatic activity of DNA methyltransferases (DNMT1) directly, except the principal benefits of being an HDAC inhibitor [67]. Panobinostat may exert the HDAC inhibitor activity to inhibit HDAC6 and HDAC7 expression. We propose that panobinostat inhibits HDAC6 and HDAC7 expression upregulation and subsequently attenuates RUNX2 expression, thereby downregulating MMP13. However, we cannot rule out the DNMT inhibition property of panobinostat. Recently, panobinostat was shown to directly inhibit the DNMT activity [1]. By contrast, it may serve as an inhibitor of DNMT to regulate the expression of HDAC4 in OA rats. However, the exact mechanisms of HDAC4 upregulation and HDAC6 and HDAC7 downregulation by panobinostat treatment require further investigation.

In conclusion, the intra-articular administration of panobinostat ameliorated OA progression and the associated nociceptive behaviors, including mechanical allodynia and weight-bearing distribution, in the ACLT-induced OA rat model. Immunohistochemical analysis and micro-CT demonstrated the inhibition of ACLT-induced OA progression after panobinostat injection. Besides, our results indicate that the modulations of HDAC4, HDAC6, and HDAC7 involve the protection of articular cartilage from degeneration by preventing chondrocyte hypertrophy. Panobinostat exerts a dual chondroprotective effect on articular cartilage degradation and chondrocyte hypertrophy by not only upregulating HDAC4, but also downregulating HDAC6 and HDAC7 by repressing RUNX2 and its downstream effector MMP13. These findings suggest that HDAC4 upregulation and HDAC6 and HDAC7 downregulation may provide chondroprotective effects in patients with early-stage OA. At present, the marine-derived HDAC inhibitor panobinostat has been approved by the US-FDA for use in clinical diseases. It could also be advantageous in future translational medicine. Thus, we believe that our findings will be valuable to develop promising therapeutic strategies for OA and other forms of arthritis.

## 4. Materials and Methods

### 4.1. Animals

Three-month-old male Wistar rats weighing 295–320 g were housed in ventilated racks on a 12-h light–dark cycle under climate-controlled conditions of 22–24 °C with a relative humidity of 60–65%. Food and water were provided ad libitum. All animals in our study lived freely without restrictions and had good appetites until sacrifice. 

### 4.2. Surgical Technique for OA Induction

OA was induced in rats via ACLT on the right knee, whereas the left knee was not treated. During the surgical procedure, rats were anesthetized with 3% isoflurane inhalation. Anesthesia was considered adequate when there was no flexor withdrawal upon a noxious foot pinch. The surgical procedure was modified from the protocol described in previous studies [68,69]. The right knee was briefly shaved and disinfected with iodine solution. A medial parapatellar incision was made in the skin, and medial arthrotomy was then performed. ACL was exposed, identified visually, and then cut through the midsubstance by a scalpel blade. Later, the anterior drawer test was performed to ensure the success of the procedure. For sham surgery, ACL was only exposed but not transected. After surgery, treated rats were not immobilized and allowed daily unrestricted cage activities. Animals were observed daily during the recovery period to confirm that wound healing progressed normally and closely monitored for infections and other complications until sacrifice.

### 4.3. Experimental Design and Intra-Articular Injection of Panobinostat

The animals were randomly allocated into the following experimental groups: Group I: ACLT group (*n* = 7); animals that underwent ACLT. Group II: ACLT + 2 µg panobinostat group (*n* = 7); animals that underwent ACLT and were injected intra-articularly with 50 µL of 2 µg panobinostat (catalog no. sc-208148, Santa Cruz, Delaware Avenue, CA, USA) were the low-dose group. Group III: ACLT + 10 µg panobinostat group (*n* = 7); animals that underwent ACLT and were injected intra-articularly with 50 µL of 10 µg panobinostat were the high-dose group. Group IV: control group (*n* = 7); rats that received no surgery and treatment. Rats in the ACLT + panobinostat groups were intra-articularly injected with panobinostat (2 or 10 µg per week) from the 6th to 17th week after ACLT, whereas the other groups received an intra-articular injection of vehicle (5% dimethyl sulfoxide in 50 µL saline). In the end, all animals were sacrificed and their knee joints were immediately collected for histopathological and micro-CT analyses. Immunohistochemical analysis was performed to examine the effect of panobinostat on the expressions of HDAC4, HDAC6, HDAC7, RUNX2, and MMP13 in articular chondrocytes.

### 4.4. Assessment of Nociception

All tests were performed during the light phase. Before the start of the test, rats were acclimated to laboratory conditions for at least 30 min. Pain-related behaviors evaluated by nociception (secondary mechanical allodynia and hind paw weight distribution) and changes in the knee joint width were blinded to group allocation and assessed weekly after ACLT.

### 4.5. Secondary Mechanical Allodynia

Mechanical allodynia was assessed by measuring the withdrawal thresholds of the ipsilateral hind paw in response to a mechanical stimulus using the calibrated von Frey filaments (North Coast Medical, Inc. Morgan Hill, CA, USA). The withdrawal threshold was determined by Chaplan’s “up–down” method, involving the use of alternate large and small fibers [70]. Each von Frey filament was applied to the plantar surface of the hind paw for a 5-s period. A positive response was defined as a rapid withdrawal of the hind paw on the application of the stimulus. Once the rat quickly lifted its paw in response to pressure, the response was noted as positive and the filament size was recorded. A weaker filament was subsequently used to test allodynia until no response occurred. A positive response with a given filament for more than three trials determined the paw withdrawal threshold.

### 4.6. Weight-Bearing Distribution

Pain behavior was measured as a weight-bearing asymmetry between the ACLT-induced OA (ipsilateral) and control (contralateral) hind paw using a dual-channel weight averager (Sigma Technology Corporation, Taipei. Taiwan), which independently measures the weight-bearing to each hind paw. In brief, rats were placed in a brown plastic chamber, and each hind paw was rested on a separated force plate. The force exerted by each hind limb (measured in g) was averaged over a 5-s period. Each data point is the average of three readings. The difference between ipsilateral and contralateral was expressed as the hind paw weight distribution [37,71]. The weight distribution of the hind paw was expressed as the difference in weight (g) between the OA-induced and contralateral control limbs.

### 4.7. Joint Width Measurement

The width of the knee joint was measured from the medial to the lateral aspects of the knee joint (at approximately the level of the medial and lateral joint lines) using a vernier caliper (Aesculap, Germany). Changes in knee joint width, a measure of knee joint inflammation, were recorded every week before and after ACLT for up to 24 weeks.

### 4.8. Sample Preparation of the Knee

At the 24th week, all rats were sacrificed by deep anesthesia with 2.5% isoflurane and perfused intracardially with 4% paraformaldehyde in 0.1 mol/L phosphate-buffered saline (PBS; containing 1% sodium nitrite and 0.2 U/mL heparin). For micro-CT scanning, the knee of the hind limbs was collected and fixed in 10% neutral formalin for 1 week at 4 °C. Knee samples were then decalcified for histopathological evaluation and immunohistochemical staining.

### 4.9. Micro-CT Imaging

To assess the three-dimension morphology and microarchitectural properties of subchondral bones, micro-CT analysis was performed by the Taiwan Mouse Clinic. For each right knee of rats, the tibia and femur were scanned and reconstructed by a microfocal CT (Skyscan 1076, Bruker, MA, USA) using an X-ray (tube voltage at 50 kV and beam current at 140 μA) and an aluminum filter of 0.5 mm. The scanning angular rotation was 180° with a rotation step of 0.8°. Each sample was exposed for 3300 ms, and the image isotropic voxel size was 9 μm. After scanning, three-dimensional images were reconstructed and analyzed with manufacturer-provided software. For the subchondral plate, the regions of interest (ROIs) were selected and analyzed with an area of 1.6 × 1.5 mm^2^. Beneath the ROI of the subchondral plate, a cuboid of trabecular bone was selected and analyzed with an area of 1.6 × 1.5 × 0.6 mm^3^. The following three-dimensional morphometric parameters were measured to describe the bone mass and structure: bone surface (mm^2^), trabecular number (Tb.N; mm^−1^), and BMD (mm^3^).

### 4.10. Histopathological Evaluation Using the OARSI Scoring System

After micro-CT imaging, the knee specimens were fixed in 10% formaldehyde for 48 h and decalcified in 12.5% PBS–ethylene diamine tetraacetic acid solution for 3 weeks. Then, the joints were sectioned midsagittally; washed with tap water; and placed in embedding cassettes for dehydration, clearing, and infiltration by an automatic tissue processor (Tissue-Tek, Sakura Finetek Japan Co. Ltd., Tokyo, Japan). Sections were stained with hematoxylin and eosin and Safranin O/Fast Green to assess the general morphology and matrix proteoglycan of the cartilage. Articular cartilage was graded under microscopic examination according to the OARSI grading system [72]. The staining sections were statistically graded by analysis under a microscope (DM 6000B, Leica Inc., Wetzlar, Germany) with an image output system (idea SPOT, Diagnostic Instruments Inc., Sterling Heights, MI, USA) [72]. All slides were evaluated by two experienced investigators who were blinded to the treatment groups.

### 4.11. Immunohistochemical Staining 

Cartilage specimens were processed for immunohistochemical analysis as per previous studies [42,64]. In brief, the paraffin-embedded sections were deparaffinized with xylene and dehydrated in a graded series of ethanol solution. Then, endogenous peroxidase activity was quenched by incubating in 0.3% hydrogen peroxide for 30 min. The antigen was retrieved by treating with proteinase K (20 mM; Sigma, St Louis, MO, USA) in PBS for 20 min. The sections were incubated with PBS containing 4% normal horse serum for 30 min to block nonspecific binding. The sections were then incubated with specific primary antibodies, including anti-HDAC4 (1:100; catalog no. ab12172; Abcam, Cambridge, UK), anti-HDAC6 (1:100; catalog no. GTX100722; GeneTex, Irvine, CA, USA), anti-HDAC7 (1:100; catalog no. ab1441; Abcam), anti-RUNX2 (1:100; catalog no. ab76956; Abcam), and anti-MMP13 (1:100; catalog no. ab39012; Abcam), overnight at 4 °C. Then, the sections were incubated for 90 min with biotinylated antirabbit immunoglobulin G (Vector Laboratories, Burlingame, CA, USA), which was diluted 1:200 with 1% bovine serum albumin in PBS. Thereafter, sections were incubated with an avidin–biotin complex using an ABC kit (Vectastain ABC kit; Vector Laboratories Inc., Burlingame, CA, USA), followed by incubation with 3,3′-diaminobenzidine tetrahydrochloride for 5 min. The different antigens present in each cartilage specimen were quantified by determining the number of chondrocytes that stained positive in the entire thickness of cartilage, as previously described [73]. Each slide was reviewed by two independent readers who were blinded to the treatment groups. Each section was analyzed by a microscope and an image output system. We acquired immunoreactive positive cells at 200 × magnification in six fields.

### 4.12. Statistical Analysis

All data are expressed as means ± standard error of the mean and analyzed using SigmaPlot version 11.0 (Systat Software, Inc., San Jose, CA, USA). The data for nociceptive behaviors; knee swelling; micro-CT analysis; and HDAC4-, HDAC6-, HDAC7-, RUNX2-, and MMP13-positive cell quantitations were analyzed by Kruskal–Wallis one-way analysis of variance after the normality test. Moreover, one-way analysis of variance followed by the Student–Newman–Keuls post-hoc test was used to analyze other data. *p*-values < 0.05 were considered statistically significant.

## Figures and Tables

**Figure 1 ijms-22-07290-f001:**
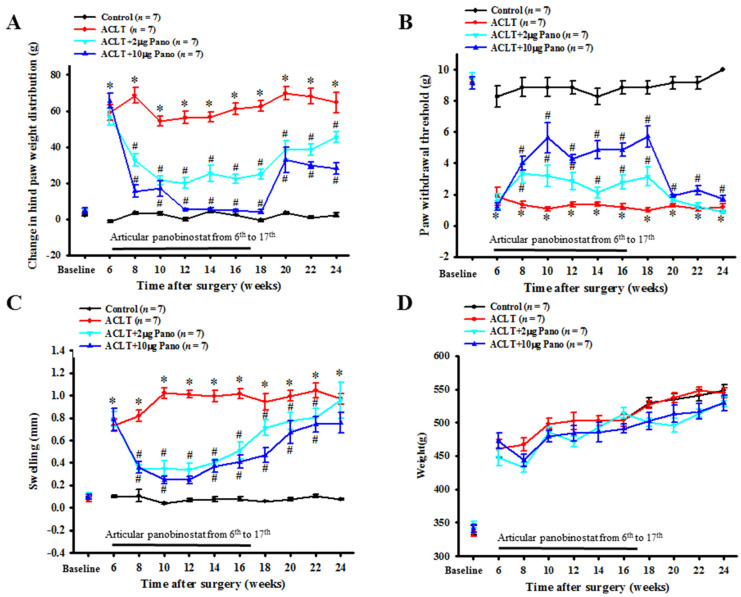
Effects of panobinostat on ACLT-induced OA model. Time courses of the effects of panobinostat on (**A**) ACLT-induced hind paw weight-bearing deficits, (**B**) mechanical allodynia, (**C**) knee swelling, and (**D**) rat weight. The rats in the control group did not receive surgery and treatment, whereas rats in the ACLT group underwent ACLT. Rats in the ACLT + panobinostat groups were intra-articularly injected with panobinostat (2 or 10 µg per week) from the 6th to 17th week after ACLT, whereas the other groups received an intra-articular injection of vehicle. Each value is presented as means ± SEM for each group. ACLT, anterior cruciate ligament transection; OA, osteoarthritis; Pano, panobinostat; SEM, standard error of the mean. (* *p* < 0.05 vs. the control group; ^#^
*p* < 0.05 vs. the ACLT group).

**Figure 2 ijms-22-07290-f002:**
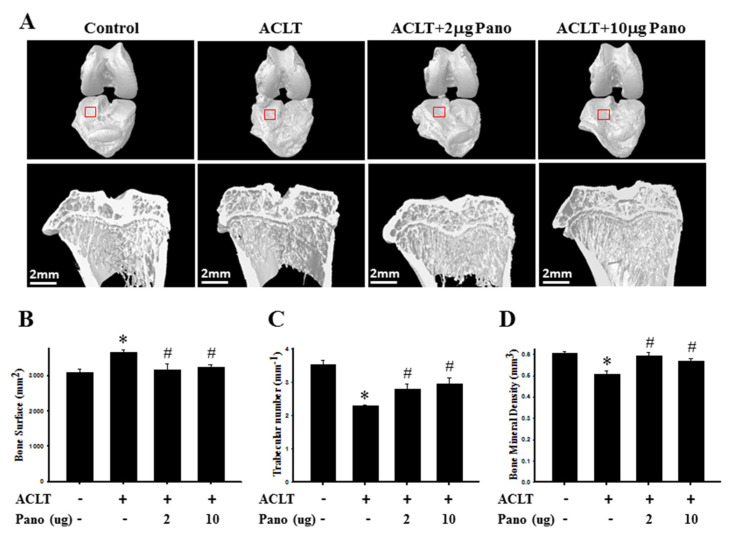
Micro-CT analysis of the bone structure in ACLT model and panobinostat treatment. (**A**) Representative three-dimensional renderings of the tibial and femoral condyles (upper panel) and sagittal views (middle panel) scanned via micro-CT. The sagittal view of micro-CT images show changes in osteoarthritic differences in the subchondral bone structures of medial femoral and tibial compartments. Red frame is the region of tibial plateau in the knee joint. (**B**) Quantitative analysis of bone surface (mm^2^), (**C**) trabecular number (mm^−1^), and (**D**) bone mineral density (mm^3^). ACLT, anterior cruciate ligament transection; Pano, panobinostat. (* *p* < 0.05 vs. the control group; ^#^
*p* < 0.05 vs. the ACLT group).

**Figure 3 ijms-22-07290-f003:**
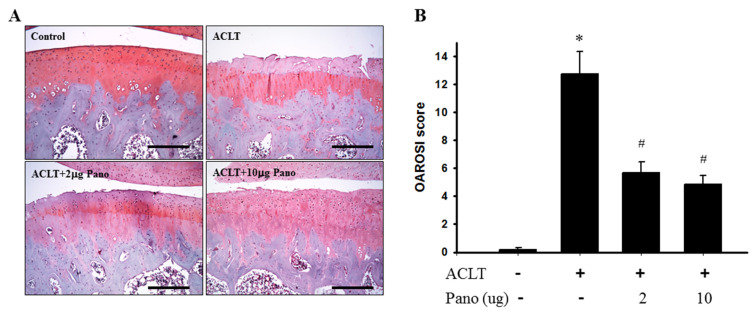
Histopathological evaluation of the tibia in knee joints after panobinostat treatment in an ACLT rat model. (**A**) Safranin O/Fast Green staining was performed on the histological sections of knee joints from the control, ACLT, and ACLT + panobinostat (2 or 10 μg) groups. Representative images of Safranin O/Fast Green staining for articular cartilage show cartilage damages in the ACLT knee compared with panobinostat treatment. The scale bar represents 250 μm. (**B**) Histopathological changes in the knee joints of the four studied groups were evaluated using the OARSI scoring system. Histogram shows the OARSI scores of the control, ACLT, and ACLT + panobinostat (2 or 10 μg) groups. OARSI, Osteoarthritis Research Society International; ACLT, anterior cruciate ligament transection; Pano, panobinostat. (* *p* < 0.05 vs. the control group; ^#^
*p* < 0.05 vs. the ACLT group).

**Figure 4 ijms-22-07290-f004:**
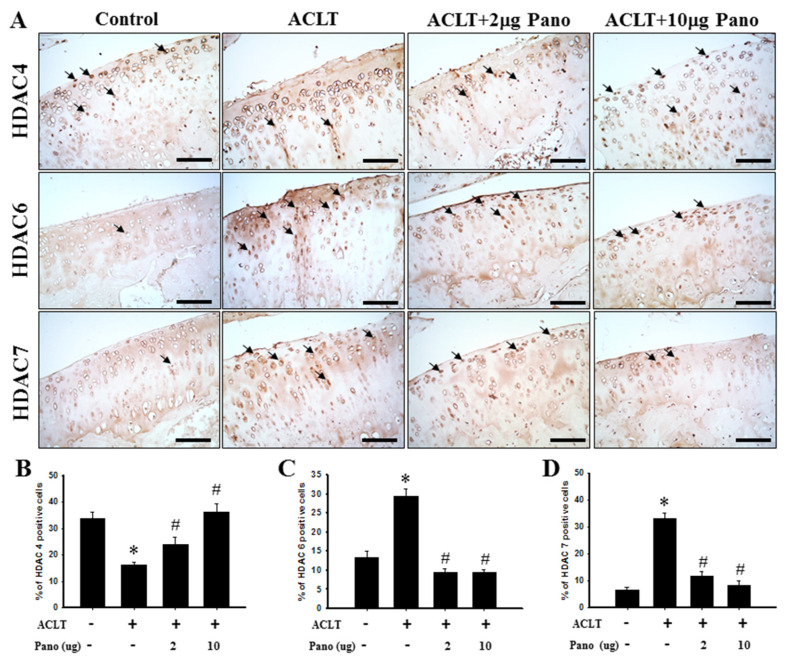
Effects of panobinostat on the expression of HDAC4, HDAC6, and HDAC7 in cartilage tissues. (**A**) Immunohistochemical staining of HDAC4, HDAC6, and HDAC7 in joint sections from the control, ACLT, ACLT + panobinostat (2 or 10 μg) groups. The immunoreactive positive cells are indicated in brown (arrows). Results from the quantitative analysis of the ratio of (**B**) HDAC4-, (**C**) HDAC6-, and (**D**) HDAC7-positive cells in joint sections are presented. Data are expressed as means ± SEM for each group. Scale bar represents 100 μm. HDAC, histone deacetylase; ACLT, anterior cruciate ligament transection; Pano, panobinostat; SEM, standard error of the mean. (* *p* < 0.05 vs. the control group; ^#^
*p* < 0.05 vs. the ACLT group).

**Figure 5 ijms-22-07290-f005:**
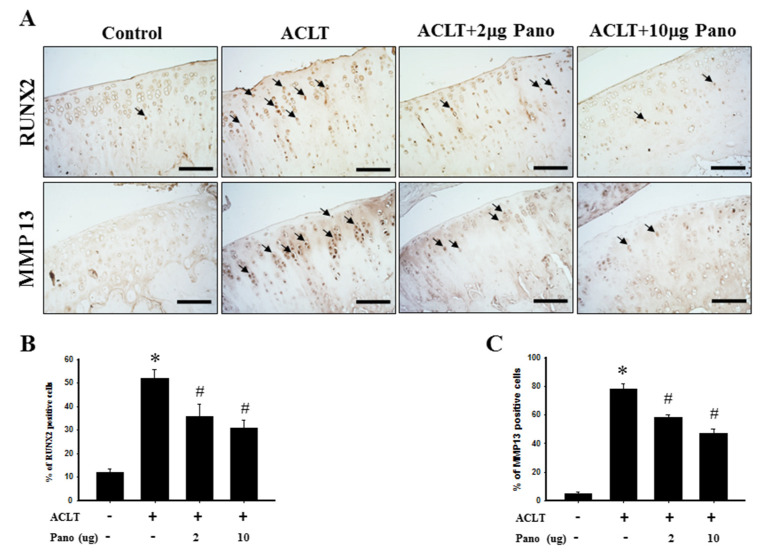
Effects of panobinostat on the expression of RUNX2 and MMP13 in cartilage tissues. (**A**) Immunohistochemical staining of RUNX2 and MMP13 in joint sections from the control, ACLT, ACLT + panobinostat (2 or 10 μg) groups. The immunoreactive positive cells are indicated in brown (arrows). Results from the quantitative analysis of the ratio of (**B**) RUNX2- and (**C**) MMP13-positive cells in joint sections are presented. Data are expressed as means ± SEM for each group. Scale bar represents 100 μm. RUNX2, runt-domain transcription factor-2; MMP13, matrix metalloproteinase-13; ACLT, anterior cruciate ligament transection; Pano, panobinostat; SEM, standard error of the mean. (* *p* < 0.05 vs. the control group; ^#^
*p* < 0.05 vs. the ACLT group).

## Data Availability

The datasets generated during and/or analyzed during the current study are available from the corresponding author on reasonable request.

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
