# Peer review of "Chondroprotective Effects of a Histone Deacetylase Inhibitor, Panobinostat, on Pain Behavior and Cartilage Degradation in Anterior Cruciate Ligament Transection-Induced Experimental Osteoarthritic Rats"

_ijms, 2021, doi:10.3390/ijms22147290_

Round 1

Reviewer 1 Report

The authors investigated the protective effects of panobinostat on pain behaviour and cartilage degradation in an ACLT model (rat). The title is not informative enough since the focus is on histone deacetylases and this is indeed interesting. This aspect should be somehow included in the title. Despite 200 words are short some crucial informations could be added to the abstract such as number of animals included concentrations of the agent tested. The introduction is long, but I enjoyed it to read about the background of this compound. In the discussion the hypothesis for how Pan. suppresses Runx and MMP13 via the particular HDACs should be explained- which signalling pathway might be involved? In this regard also the effects on bone require a hypothesis Lines 344-360. The activity of the MMP is not investigated. Some English adaptions should be done.

Line 47: „treating“ is redundant

Line 54: rather „has“ instead of „was“?

Line 56: why is reference no 8 blue?

Line 64: please rewrite this sentence „Since…“ and also the sentence in line 106: „It is known…“

Line 119: „derived“ sounds surplus

Line 151: weight bearing distribution should be explained in the beginning. Distribution of body weight between both hind legs?

Line 166: „in the width“ could be omitted

Line 198: write „rougher and more irregular“…

Fig. 3: which part oft he knee joint cartilage is shown (tibial or femoral)?

Line 287: write „higher in number“

Figure 5. Is the active enzyme labeled (MMP13)?

Line 313: „weight-bearing distribution defect“ this statement sounds strange (style)

Line 320: „MMP13 signalling“ what means MMP13 signaling? It is an enzyme with no downstream signaling pathway.

Line 329: synovial membrane hyperplasia – was the degree of synovitis investigated?

Line 375: „suppresses… hypertrophy related genes…“

Line 394: substitute „than“ by „compared to“

A clear hypothesis should be expressed – how is pain suppressed?

4.1. provide already here the animal number

Line 435: signs of illness“ sounds contradictory to less pain

Line 533: write „stained sections“, were the observer blinded, how many observers were involved?

Author Response

Response to Comments from Reviewer # 1

To Reviewer # 1:

The authors investigated the protective effects of panobinostat on pain behaviour and cartilage degradation in an ACLT model (rat). The title is not informative enough since the focus is on histone deacetylases and this is indeed interesting. This aspect should be somehow included in the title. Despite 200 words are short some crucial informations could be added to the abstract such as number of animals included concentrations of the agent tested.

Our reply: We thank the reviewer’s comments and suggestions. The point-to-point responses to the comments are listed below.

  1. The title is not informative enough since the focus is on histone deacetylases and this is indeed interesting.

Our reply: We resubmitted a revised manuscript entitled ” Chondroprotective effects of a histone deacetylase inhibitor, panobinostat, on pain behavior and cartilage degradation in anterior cruciate ligament transection-induced experimental osteoarthritic rats”.

  1. Despite 200 words are short some crucial informations could be added to the abstract such as number of animals included concentrations of the agent tested.

Our reply: As suggested by the reviewer, we modified the sentence with some words as follows: Intra-articular administration of panobinostat 2 or 10 mg (each group, n=6) per week from the 6th to 17th week attenuates ACLT-induced nociceptive behaviors, such as secondary mechanical allodynia and weight-bearing distribution.

  1. The introduction is long, but I enjoyed it to read about the background of this compound.

Our reply: Thanks for the comment. We have tried hard to downsize the INTRODUCTION section by about one fourth.

  1. In the discussion the hypothesis for how Pan. suppresses Runx and MMP13 via the particular HDACs should be explained- which signalling pathway might be involved?

Our reply: Thanks for the comment. We have explained our hypothesis as mentioned in DISCUSSION section as follows:

Panobinostat may exert HDACi activity to inhibit the expression of HDAC6 and HDAC7. We proposed that panobinostat may inhibit the upregulation of HDAC6 and HDAC7 expression and subsequently attenuate the RUNX2 expression results in a downregulation of MMP13. We cannot rule out the DNA methyltransferase (DNMT) inhibition property of panobinostat. Recently, it has been confirmed that panobinostat can directly inhibit DNMT activity [1], and it may serve as an inhibitor of DNMT to regulate the expression of HDAC4 in OA rats. However, the exact mechanisms of panobinostat treatment on upregulated HDAC4 and downregulated HDAC6 and HDAC7 require further investigation.

Reference:

  1. Juárez-Mercado, K. E.; Prieto-Martínez, F. D., Expanding the Structural Diversity of DNA Methyltransferase Inhibitors. 2020, 14, (1).

  1. In this regard also the effects on bone require a hypothesis Lines 344-360.

Our reply: Thank you for your comment. Cumulative studies support the concept of the progressive loss of articular cartilage and the acceleration of subchondral bone turnover accompanied by microarchitecture changes in the subchondral trabecular bone of OA joints, which is characterized by increased subchondral plate thickness and the formation of osteophytes [1,2]. Based on our results, we hypothesized that panobinostat might indirectly reduce cartilage degradation by protecting subchondral bone from resorption.

References:

  1. Yu, D.; Xu, J.; Liu, F.; Wang, X.; Mao, Y.; Zhu, Z., Subchondral bone changes and the impacts on joint pain and articular cartilage degeneration in osteoarthritis. Clinical and experimental rheumatology 2016, 34, (5), 929-934.
  2. Li, G.; Yin, J.; Gao, J.; Cheng, T. S.; Pavlos, N. J.; Zhang, C.; Zheng, M. H., Subchondral bone in osteoarthritis: insight into risk factors and microstructural changes. Arthritis research & therapy 2013, 15, (6), 223.

  1. The activity of the MMP is not investigated.

Our reply: Thank you for your comment. MMP13 is the enzyme responsible for degeneration of the cartilage extracellular matrix. It is known that the upregulation of MMP13 protein expression plays a crucial role in the pathogenesis of OA (1-3). Several previous studies have clearly demonstrated the upregulation of MMP13 protein expression in OA rats by immunohistochemistry staining (4-8). The above studies also investigate the role of MMP13 protein expression on possible mechanisms in compounds or bioactive substances that protect OA progression. According to previous studies, we used the immunohistochemistry technique to evaluate MMP13 protein expression in the present study. Thank you for your comment. Based on your suggestion, we will consider performing an MMP13 activity assay in future studies.

References:

  1. Wang, M.; Sampson, E. R.; Jin, H.; Li, J.; Ke, Q. H.; Im, H. J.; Chen, D., MMP13 is a critical target gene during the progression of osteoarthritis. Arthritis research & therapy 2013, 15, (1), R5.
  2. Hu, Q.; Ecker, M., Overview of MMP-13 as a Promising Target for the Treatment of Osteoarthritis. 2021, 22, (4).
  3. Kwon, J. Y.; Lee, S. H.; Na, H. S.; Jung, K.; Choi, J.; Cho, K. H.; Lee, C. Y.; Kim, S. J.; Park, S. H.; Shin, D. Y.; Cho, M. L., Kartogenin inhibits pain behavior, chondrocyte inflammation, and attenuates osteoarthritis progression in mice through induction of IL-10. Scientific reports 2018, 8, (1), 13832.
  4. Li, H.; Wang, D.; Yuan, Y.; Min, J., New insights on the MMP-13 regulatory network in the pathogenesis of early osteoarthritis. Arthritis research & therapy 2017, 19, (1), 248
  5. Hu, J.; Zhou, J.; Wu, J.; Chen, Q.; Du, W.; Fu, F.; Yu, H.; Yao, S.; Jin, H.; Tong, P.; Chen, D.; Wu, C.; Ruan, H., Loganin ameliorates cartilage degeneration and osteoarthritis development in an osteoarthritis mouse model through inhibition of NF-κB activity and pyroptosis in chondrocytes. Journal of ethnopharmacology 2020, 247, 112261.
  6. Hu, X.; Ji, X.; Yang, M.; Fan, S.; Wang, J.; Lu, M.; Shi, W.; Mei, L.; Xu, C.; Fan, X.; Hussain, M.; Du, J.; Wu, J.; Wu, X., Cdc42 Is Essential for Both Articular Cartilage Degeneration and Subchondral Bone Deterioration in Experimental Osteoarthritis. Journal of bone and mineral research : the official journal of the American Society for Bone and Mineral Research 2018, 33, (5), 945-958.
  7. Zhang, L.; Hu, H.; Tian, F.; Song, H.; Zhang, Y., Enhancement of subchondral bone quality by alendronate administration for the reduction of cartilage degeneration in the early phase of experimental osteoarthritis. Clinical and experimental medicine 2011, 11, (4), 235-43.
  8. Gu, X. D.; Wei, L.; Li, P. C.; Che, X. D.; Zhao, R. P.; Han, P. F.; Lu, J. G.; Wei, X. C., Adenovirus-mediated transduction with Histone Deacetylase 4 ameliorates disease progression in an osteoarthritis rat model. Int Immunopharmacol 2019, 75, 105752.

  1. Some English adaptions should be done.

Our reply: Thanks for the comment. The revised manuscript has been proofread and corrected by a professional English editor as follows:

  1. Line 47: „treating“ is redundant

Our reply: We have omitted the word already in our revised manuscript.

  1. Line 54: rather „has“ instead of „was“?

Our reply: Thanks for pointing out this error. The revised manuscript has rewritten in the frist paragraph of the INTRODUCTION section as follows:

Panobinostat (LBH589), a histone deacetylase (HDAC) inhibitor, was approved for treating patients with multiple myeloma and other hematological malignancies by the United States Food and Drug Administration (US-FDA) in 2015 [1]. LBH589 was developed from a marine natural product, psammaplin A (PSA) [2-4], which was first isolated from a marine sponge Psammaplvsilla sp. in 1987 [5]. At first, PSA was discovered as an HDAC inhibitor for anticancer properties from p21, a cyclin-dependent kinase 2 promoter assay system [6, 7]. Because of the weak physiological stability of PSA, medicinal chemists attempted to improve its chemical properties [8]. Although several interesting derivatives were developed, the pharmacological profile of PSA was not improved successfully [7, 8]. Subsequently, another natural compound, trichostatin A (also an HDAC inhibitor), was isolated using the same p21 promoter assay [7-9]. LAK974 was synthesized based on the analysis of a two-dimensional pharmacophore model of trichostatin A; however, LAK974 showed significant activity in vitro but poor efficacy in vivo [7]. Structure modification was continued by computational docking studies to obtain LAQ824, which showed increased antiproliferative effects in several cancer cell lines (including A549 [lung], HCT166 [colon], and MDA435 [breast]) and decreased apoptotic death in normal human dermal fibroblasts [7]. Because the safety of LAQ824 in preclinical development was unclear and it induced the body weight loss problem in vivo [7], LBH589 was finally optimized by further synthetic design and showed a significant tumor regression effect in HCT116 xenograft model with minimal body weight loss [7, 8].

  1. Line 56: why is reference no 8 blue?

Our reply: Thanks for pointing out this error. Our reply is similar to comment no. 9.

  1. Line 64: please rewrite this sentence „Since…“ and also the sentence in line 106: „It is known…“

Our Reply: Thanks for the comments. We have rewritten this sentence (Line 64) , as similar to comment no. 9. Also, we have removed and rewritten the sentence (line 106) in the DISSCUSION, as following: Chondrocytes respond to the accumulation of injurious biochemical and biomechanical insults resulted in hypertrophy-like phenotype, which plays a vital role in initiation and progression of OA.

  1. Line 119: „derived“ sounds surplus

Our reply: We have rewritten the sentence already in our revised manuscript as follows: A study showed that RUNX2 deletion decelerates OA progression in an experimental OA model using medial meniscal destabilization surgery

  1. Line 151: weight bearing distribution should be explained in the beginning. Distribution of body weight between both hind legs?

Our reply: Thanks a lot for the reviewer’s comment. Duo to the format in this journal, we explained in more detail in MATERIALS and METHODS section.

  1. Line 166: „in the width“ could be omitted

Our reply: We have rewort the sentcent already in our revised manuscript as foolows: As shown in Figure 1C, the ACLT group experienced a significantly increased swelling of the hind limb knee joint before panobinostat treatment.

  1. Line 198: write „rougher and more irregular“…

Our reply: Thanks for reviewer’s suggestion. We have replaced the words already in our revised manuscript.

  1. Fig. 3: which part of the knee joint cartilage is shown (tibial or femoral)?

Our reply: Thanks for the comments. In our revised manuscript, we showed that the part of the knee joint cartilage is tibial in Fig. 3.

  1. Line 287: write „higher in number“

Our reply: Thanks for reviewer’s suggestion. We have added the words already in our revised manuscript.

  1. Figure 5. Is the active enzyme labeled (MMP13)?

Our reply: Base on several previous studies, we only assay the MMP-13 protein expression by immunohistochemical staining. Our reply is similar to comment no. 6.

  1. Line 313: „weight-bearing distribution defect“ this statement sounds strange (style)

Our reply: Thanks for pointing out this error. We have omitted the word, “defect”, already in our revised manuscript.

  1. Line 320: „MMP13 signalling“ what means MMP13 signaling? It is an enzyme with no downstream signaling pathway.

Our reply: Thanks for pointing out this error. We have omitted the word, “signalling”, already in our revised manuscript.

  1. Line 329: synovial membrane hyperplasia – was the degree of synovitis investigated?

Our reply: Thanks for the comments. We did not investigate the degree of synovitis in our study. Alternatively, we focused on evaluating the effects of panobinostat on the subchondral bone quality and cartilage degradation.

  1. Line 375: „suppresses… hypertrophy related genes…“

Our reply: Thanks for pointing out this error. We have replaced the words already in our revised manuscript.

  1. Line 394: substitute „than“ by „compared to“

Our reply: Thanks for pointing out this error. The error has been corrected already in our revised manuscript.

  1. A clear hypothesis should be expressed – how is pain suppressed?

Our reply: Thanks for the valuable comments. We have provided our hypothesis as mentioned in DISCUSSION section as follows:

HDAC inhibitors are the most common used as anti-cancer drugs. However, several studies have indicated the therapeutic potential of HDAC inhibitors as anti-inflammatory and immunosuppressive agents [1-3] as well as effective analgesics [4]. OA is a type of degenerative joint disease accompanied with cartilage degeneration and ongoing nociception. Pathological inflammation plays a vital role in the development and progression of cartilage degeneration and nociceptive sensitization in OA [5]. Moreover, RUNX2 and MMP13 are involved in inflammation and destruction in OA [6-8]. Both cartilage degradation and nociceptive sensitization can be relieved by MMP13 inhibition [2, 9, 10]. MMP13 is specifically expressed in the cartilage of human OA patients but is not present in normal adult cartilage [11, 12]. In the present result, the HDAC inhibitor panobinostat significantly inhibited ACLT-induced RUNX2 and MMP13 protein expression (Figure 5). Therefore, we suggest that panobinostat be considered as having therapeutic value for OA via the anti-inflammatory properties of MMP13 inhibition.

References:

  1. Cantley, M. D.; Haynes, D. R., Epigenetic regulation of inflammation: progressing from broad acting histone deacetylase (HDAC) inhibitors to targeting specific HDACs. Inflammopharmacology 2013, 21, (4), 301-7.
  2. Licciardi, P. V.; Ververis, K.; Tang, M. L.; El-Osta, A.; Karagiannis, T. C., Immunomodulatory effects of histone deacetylase inhibitors. Current molecular medicine 2013, 13, (4), 640-7.
  3. Zhang, H.; Dai, X.; Qi, Y., Histone Deacetylases Inhibitors in the Treatment of Retinal Degenerative Diseases: Overview and Perspectives. 2015, 2015, 250812.
  4. Wang, W.; Cui, S. S.; Lu, R.; Zhang, H., Is there any therapeutic value for the use of histone deacetylase inhibitors for chronic pain? Brain research bulletin 2016, 125, 44-52.
  5. Schaible, H. G., Mechanisms of chronic pain in osteoarthritis. Current rheumatology reports 2012, 14, (6), 549-56.
  6. Luan, J.; Tao, H.; Su, Y., Taladegib controls early chondrocyte hypertrophy via inhibiting smoothened/Gli1 pathway. Am J Transl Res 2020, 12, (5), 1985-1993.
  7. Kamekura, S.; Kawasaki, Y.; Hoshi, K.; Shimoaka, T.; Chikuda, H.; Maruyama, Z.; Komori, T.; Sato, S.; Takeda, S.; Karsenty, G.; Nakamura, K.; Chung, U. I.; Kawaguchi, H., Contribution of runt-related transcription factor 2 to the pathogenesis of osteoarthritis in mice after induction of knee joint instability. Arthritis and rheumatism 2006, 54, (8), 2462-70.
  8. Wei, F.; Zhou, J.; Wei, X.; Zhang, J.; Fleming, B. C.; Terek, R.; Pei, M.; Chen, Q.; Liu, T.; Wei, L., Activation of Indian hedgehog promotes chondrocyte hypertrophy and upregulation of MMP-13 in human osteoarthritic cartilage. Osteoarthritis and cartilage 2012, 20, (7), 755-63.
  9. Billinghurst, R. C.; Dahlberg, L.; Ionescu, M.; Reiner, A.; Bourne, R.; Rorabeck, C.; Mitchell, P.; Hambor, J.; Diekmann, O.; Tschesche, H.; Chen, J.; Van Wart, H.; Poole, A. R., Enhanced cleavage of type II collagen by collagenases in osteoarthritic articular cartilage. The Journal of clinical investigation 1997, 99, (7), 1534-45.
  10. Baragi, V. M.; Becher, G.; Bendele, A. M.; Biesinger, R.; Bluhm, H.; Boer, J.; Deng, H.; Dodd, R.; Essers, M.; Feuerstein, T.; Gallagher, B. M., Jr.; Gege, C.; Hochgürtel, M.; Hofmann, M.; Jaworski, A.; Jin, L.; Kiely, A.; Korniski, B.; Kroth, H.; Nix, D.; Nolte, B.; Piecha, D.; Powers, T. S.; Richter, F.; Schneider, M.; Steeneck, C.; Sucholeiki, I.; Taveras, A.; Timmermann, A.; Van Veldhuizen, J.; Weik, J.; Wu, X.; Xia, B., A new class of potent matrix metalloproteinase 13 inhibitors for potential treatment of osteoarthritis: Evidence of histologic and clinical efficacy without musculoskeletal toxicity in rat models. Arthritis and rheumatism 2009, 60, (7), 2008-18.
  11. Mitchell, P. G.; Magna, H. A.; Reeves, L. M.; Lopresti-Morrow, L. L.; Yocum, S. A.; Rosner, P. J.; Geoghegan, K. F.; Hambor, J. E., Cloning, expression, and type II collagenolytic activity of matrix metalloproteinase-13 from human osteoarthritic cartilage. The Journal of clinical investigation 1996, 97, (3), 761-8.
  12. Freemont, A. J.; Byers, R. J.; Taiwo, Y. O.; Hoyland, J. A., In situ zymographic localisation of type II collagen degrading activity in osteoarthritic human articular cartilage. Annals of the rheumatic diseases 1999, 58, (6), 357-65.

  1. 4.1. provide already here the animal number

Our reply: Thanks for the comments. We have provided the number of animal (n=7) in each group as described in 4.3 Experimental design and intra-articular injection of panobinostat.

  1. Line 435: signs of illness“ sounds contradictory to less pain

Our reply: Thanks for reviewer’s suggestion. We have rewritten the sentence as follows: All animals in our study lived freely without restrictions and had good appetites until sacrifice.

  1. Line 533: write „stained sections“, were the observer blinded, how many observers were involved?

Our reply: Thanks for the comments. In our revised manuscript, we have added the sentence in 4.10 Histopathological evaluation using the OARSI scoring system as follows: All slides were evaluated by two experienced investigators who were blinded to the treatment groups

Reviewer 2 Report

The manuscript entitled “The chondroprotective effects of panobinostat on pain behavior and cartilage degradation in anterior cruciate ligament transection-induced experimental osteoarthritic rats” is an interesting study. However, I have several comments.

In general, English should be improved in the introduction, methods and result sections. There are several sentences unclear. For example, lines 51-55: “Initially, PSA was discovered as an inducer of p21 gene expression, a cyclin-dependent kinase 2 (CDK2) inhibitor, indicating that PSA was a new hit for inducing p21 expression to inhibit the characterized protein, HDAC [6], its 53 potential anti-cancer effect with inhibition function of HDAC was further gained great attention [7-9].” The sentence should be rewritten.

Line 56 and line 84: “[[10]8].”and “(10)”  should be corrected.

Line 64: it is not clear which cancer cell line and normal cell line.

Line 67: Could the authors specify the tumor?

Lines 88-90: the introduction on OA should be improved. OA is characterized not only by cartilage degeneration, subchondral bone remodeling and synovial inflammation but also by inflammation and fibrosis of the infrapatellar fat pad and meniscal damage/degeneration (DOI: 10.3390/ijms21176016; DOI: 10.1038/nrrheum.2012.69).

The quality of the figure 1 should be improved.  

Line 141, line 151, 152: “ug” should be corrected. The authors should check and correct “ug” throughout the manuscript and figures.

Line 242: “um” should be corrected.

The software used for the statistical analysis should be reported.

Author Response

Response to Comments from Reviewers

To Reviewer # 2:

The manuscript entitled “The chondroprotective effects of panobinostat on pain behavior and cartilage degradation in anterior cruciate ligament transection-induced experimental osteoarthritic rats” is an interesting study. However, I have several comments

  1. In general, English should be improved in the introduction, methods and result sections. There are several sentences unclear. For example, lines 51-55: “Initially, PSA was discovered as an inducer of p21 gene expression, a cyclin-dependent kinase 2 (CDK2) inhibitor, indicating that PSA was a new hit for inducing p21 expression to inhibit the characterized protein, HDAC [6], its 53 potential anti-cancer effect with inhibition function of HDAC was further gained great attention [7-9].” The sentence should be rewritten.

Our reply: Thanks for the comment. The revised manuscript has been proofread and corrected by a professional English editor as follows:

The revised manuscript has rewritten in the sentence (lines 51-55) as follows: At first, PSA was discovered as an HDAC inhibitor for anticancer properties from p21, a cyclin-dependent kinase 2 promoter assay system [6, 7]. Because of the weak physiological stability of PSA, medicinal chemists attempted to improve its chemical properties [8].

  1. Line 56 and line 84: “[[10]8].”and “(10)”  should be corrected.

Our reply: Thanks for the comment. The revised manuscript has corrected two errors.

  1. Line 64: it is not clear which cancer cell line and normal cell line.

Our reply: Thanks for the reviewer’s comment. We have rewritten the sentence as follows: Structure modification was continued by computational docking studies to obtain LAQ824, which showed increased antiproliferative effects in several cancer cell lines (including A549 [lung], HCT166 [colon], and MDA435 [breast]) and decreased apoptotic death in normal human dermal fibroblasts.

  1. Line 67: Could the authors specify the tumor?

Our reply: Thanks for the reviewer’s comment. We have rewritten the sentence as follows: LBH589 was finally optimized by further synthetic design and showed a significant tumor regression effect in HCT116 xenograft model with minimal body weight loss.

  1. Lines 88-90: the introduction on OA should be improved. OA is characterized not only by cartilage degeneration, subchondral bone remodeling and synovial inflammation but also by inflammation and fibrosis of the infrapatellar fat pad and meniscal damage/degeneration (DOI: 10.3390/ijms21176016; DOI: 10.1038/nrrheum.2012.69).

Our reply: Thanks a lot for reviewer’s useful suggestion. We have reorganized the sentence and cited the two references in our revised manuscript as follows: OA is a common age-related degenerative joint disease that causes pain and disability in older people.It is characterized not only by cartilage degradation, subchondral bone remodeling, and synovial inflammation but also by inflammation, infrapatellar fat pad fibrosis, and meniscal damage/degeneration.

  1. The quality of the figure 1 should be improved.

Our Reply: Thanks for the reviewer’s comment. We have provided clearer and higher quality figure 1 in our revised manuscript.

  1. Line 141, line 151, 152: “ug” should be corrected. The authors should check and correct “ug” throughout the manuscript and figures

Our reply: Thanks for pointing out some errors. We have corrected the unit as “μg” throughout the manuscript and figures.

  1. Line 242: “um” should be corrected.

Our reply: Thanks for pointing out this error. We have corrected the unit as “μm” in our revised manuscript. 

  1. The software used for the statistical analysis should be reported.

Our reply: Thanks for the reviewer’s comment. The statistical software used in our study was SigmaPlot version 11.0 (Systat Software, Inc., San Jose, CA, USA) as shown in the section of 4.12 Statistical analysis.

Round 2

Reviewer 1 Report

The sequence of the sentence: "ACLT group showed a rougher and more irregular rougher and more irregular surface" sounds a bit confusing.

line 207: This sentence should be checked concerning grammar:

line 331-332: Chondrocytes respond to the accumulation of injurious biochemical and biomechanical insults resulted in hypertrophy-like phenotype, which plays a vital role in initiation and progression of OA.

line 603: there is a surplus bracket "SigmaPlot version 11.0 ((Systat Software, Inc., San Jose, CA, USA)."

line 611: remove the surplus blank "and MOST 105-2325-B-110-001 )"

Author Response

Response to Comments from Reviewers

To Reviewer # 1:

line 207: This sentence should be checked concerning grammar:

The knee joints of the ACLT group showed a rougher and more irregular rougher and more irregular surface than those of the control rats (Fig. 2A).

Our reply: Thanks for the comment. We have rewritten the sentence already in our revised manuscript as follows:

The knee joints of the ACLT group showed a rougher and more irregular surface than those of the control rats (Fig. 2A).

line 331-332: Chondrocytes respond to the accumulation of injurious biochemical and biomechanical insults resulted in hypertrophy-like phenotype, which plays a vital role in initiation and progression of OA.

 Our reply: Thanks for the comment. We have rewritten the sentence already in our revised manuscript as follows:

It is known that chondrocytes respond to the accumulation of injurious biochemicals and biomechanical insults acquired from a hypertrophy-like phenotype, which plays a vital role in the onset and development of OA.

line 603: there is a surplus bracket "SigmaPlot version 11.0 ((Systat Software, Inc., San Jose, CA, USA)."

Our reply: Thanks for pointing out this error. We had corrected it already.

line 611: remove the surplus blank "and MOST 105-2325-B-110-001 )"

Our reply: Thanks for your careful review, we had corrected it already.